# A Tale of Two Cancers: A Current Concise Overview of Breast and Prostate Cancer

**DOI:** 10.3390/cancers14122954

**Published:** 2022-06-15

**Authors:** Franklyn De Silva, Jane Alcorn

**Affiliations:** Drug Discovery & Development Research Group, College of Pharmacy and Nutrition, 104 Clinic Place, Health Sciences Building, University of Saskatchewan, Saskatoon, SK S7N 2Z4, Canada

**Keywords:** breast cancer, prostate cancer, risk factors, cancer statistics, cancer classification, female breast anatomy, prostate anatomy, cancer pathophysiology, cancer diagnosis, cancer treatment, cancer heterogeneity, neuroendocrine cancer, drug-tolerant persister cells, multidrug resistance

## Abstract

**Simple Summary:**

Breast and prostate cancers are serious public health issues that create considerable burden to both people and healthcare systems worldwide. Cancer is a heterogeneous disease influenced by numerous components, and its diverse intricate pathology challenges disease prevention, diagnosis, treatment, and survival. Although recent statistics suggest improvements in cancer diagnosis and treatment, many challenges remain before cancers are curable. This review presents relevant summarized information related to breast and prostate cancer.

**Abstract:**

Cancer is a global issue, and it is expected to have a major impact on our continuing global health crisis. As populations age, we see an increased incidence in cancer rates, but considerable variation is observed in survival rates across different geographical regions and cancer types. Both breast and prostate cancer are leading causes of morbidity and mortality worldwide. Although cancer statistics indicate improvements in some areas of breast and prostate cancer prevention, diagnosis, and treatment, such statistics clearly convey the need for improvements in our understanding of the disease, risk factors, and interventions to improve life span and quality of life for all patients, and hopefully to effect a cure for people living in developed and developing countries. This concise review compiles the current information on statistics, pathophysiology, risk factors, and treatments associated with breast and prostate cancer.

## 1. Introduction

The worldwide burden of cancer incidence and mortality is considerable and is projected to increase despite the progress made in cancer diagnosis, treatment, and management. Of the well over 200 different types of human cancers, breast cancer (BC) and prostate cancer (PC) are among the top cancers owing to their unique features encompassing their origin, acquired mutations, gene expression patterns, modified transcriptional and signaling networks, metabolic activities, the impact of their microenvironments, and host immune responses and overall health (e.g., infections) [1,2,3,4,5,6,7,8]. In this short overview, we aim to compile a brief review of the current global cancer burden, the pathological complexity, risk factors, and treatment options for BC and PC, and some future perspectives. Further, this timely, and relevant overview of the literature has educational value for academics and clinicians.

## 2. Cancer Burden

World Health Organization (WHO) estimates indicate that cancer is among the top two causes of death in 112 of 183 countries [9,10]. In an additional 23 countries, it ranks third or fourth [9,10]. In 2020, there was 19.3 million new cancer cases (10.1 million males vs. 9.2 million females), plus 9.9 million cancer deaths (5.5 million males vs. 4.4 million females) worldwide [10,11,12,13]. Almost 50% of cases and over 50% of cancer deaths occurred in Asia, a continent with 60% of the global population [9,10,14]. In contrast, Europe had 22.8% of total cases and 19.6% of deaths, and the Americas had 20.9% cases and 14.2% mortality, while the proportion of deaths in Asia (58.3%) and Africa (7.2%) were higher than the reported incidence (49.3% and 5.7% respectively), owing to the contrasting regional distribution of cancer types and higher fatality rates [9,10,14]. The commonly diagnosed cancer and the prominent cause of death varies across countries and within each country [9]. The 2018 report by the International Agency for Research on Cancer (IARC) identified that the most frequently diagnosed and leading cause of cancer-related deaths was lung cancer [9]. However, by 2020 female BC surpassed lung cancer with approximately 2.3 million new cases (11.7%), compared to lung (11.4%), colorectal (10.0%), prostate (7.3%), and stomach (5.6%) [10,13].

GLOBOCAN 2020 reported deaths for female-BC and PC as 6.9% and 3.8% respectively [10]. Accounting for all cancer sites, there was a cumulative risk of 22.6% (males) and 18.6% (females) for incidence, and a cumulative risk of 12.6% (males) and 8.9% (females) for mortality for both sexes (ages 0–74 years), globally (24 world areas) in 2020 [10]. Detailed statistics on trends in cancer incidence and death can be found in the GLOBOCAN reports [9,10,14,15]. Globally, 1 in 5 men and 1 in 6 women will develop cancer, and 1 in 8 men and 1 in 10 women will die from cancer before age 75 [9]. An approximately 47% increase in cancer cases is projected by 2040 worldwide compared to 2020 [10]. Although incidence rates vary widely across regions, all cancer’s combined incidence rate was 19% higher in men in comparison to women in 2020 [10]. For both men and women, the incidence rate positively correlates with increasing human development index (HDI) level [10], with mortality rates around 2-fold greater in higher HDI countries relative to lower HDI countries in men [10]. Interestingly, among women, little variation exists across HDI levels [10] (Figure 1 and Figure 2).

**Figure 1 cancers-14-02954-f001:**
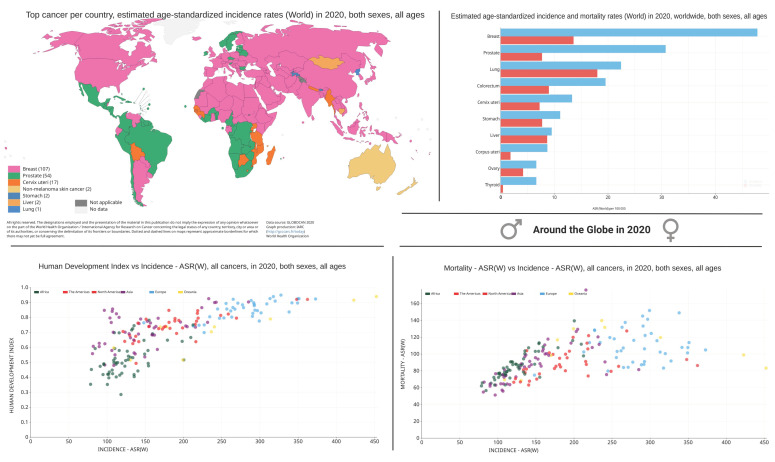
A snapshot of worldwide cancer incidence and mortality: Representation of the major cancer types (color coded) in each country based on the age standardized incidence rates for all ages and sex (world map). On the right, the bar graph represents the age standardized rates (ASR) based estimates on incidence and mortality, accounting for all ages for both male and female combined. Bottom left dot plot represents the ASR estimates on the world human development index (HDI) vs. incidence. Bottom right graph represents the global ASR estimates on the incidence vs mortality. The incidence, mortality, and prevalence information presented on ‘CANCER TODAY’ (World Health Organization) represents a collection of 36 specific types of cancer and 185 countries or territories of the world in 2020 (GLOBOCAN project, International Agency for Research on Cancer (IARC)), obtained from many cancer sites from across the globe. Figures taken from IARC website (information publicly available online accessed in 20 November 2021, https://gco.iarc.fr/today/home) [9,14,15].

BC is a common form of malignancy among women in the developing and developed world [12,16,17]. Incidence rates continue to rise in countries of all income levels [9,10,18,19]. However, there is heterogeneity in disease burden across countries of different income levels [19] and BC is the leading cause of cancer death in women [10,20]. According to 2021 estimates, on average, 76 women will be diagnosed and 15 women will die from BC every day in Canada [21]. This review will focus on female BC and readers are encouraged to access other articles on male BC [22,23,24]. North America, New Zealand, Australia, and Europe (northern and western) have the highest BC incidence in the world [25], but mortality in Europe and North America has decreased [17]. Although developing countries report lower BC incidence rates, almost 58% of worldwide deaths occur in developing nations [16], and BC incidence is increasing in Africa, Asia, and South America owing to factors such as early detection, efficient systemic therapies, initiated screening programs, and lifestyle changes [17]. As well, within these regions, mortality is increasing possibly related to the shortage of accessible state-of-the-art diagnosis and therapeutic interventions [17]. Reported survival rates range from above 80% in developed nations to under 40% in low-income nations [26] (Figure 1).

PC is the second most frequent and fifth major cause of cancer-associated death among men in 2020 [10]. PC attributes to 1.4 million of the total new cancer cases in men and 375,000 of the overall male cancer deaths worldwide [9,10,12,27,28]. In 112 countries, PC is the most frequently diagnosed cancer in males (50% of all the countries) [10]. Even though mortality rates are less variable, the incidence rates in transitioned countries are 3-fold higher than that of transitioning countries [10]. Northern and Western Europe, Australia/New Zealand, Northern America, Southern Africa, and the Caribbean have the highest incidence rates, and Asia and Northern Africa have the lowest [10]. PC is the third leading cause of death from cancer in Canada with an average of 12 lives per day [29]. Recent estimates indicate ~24,000 new PC diagnoses (20% of all new cases), and 4500 deaths (10% of cancer deaths) are expected yearly in Canadian men [30], with 1 of 8 men diagnosed with PC during his lifetime, and 1 of 29 men dying of PC in Canada [30]. PC is rare before the age of 40, and the average age at the time of diagnosis is 66 in North America [31] (Figure 2).

**Figure 2 cancers-14-02954-f002:**
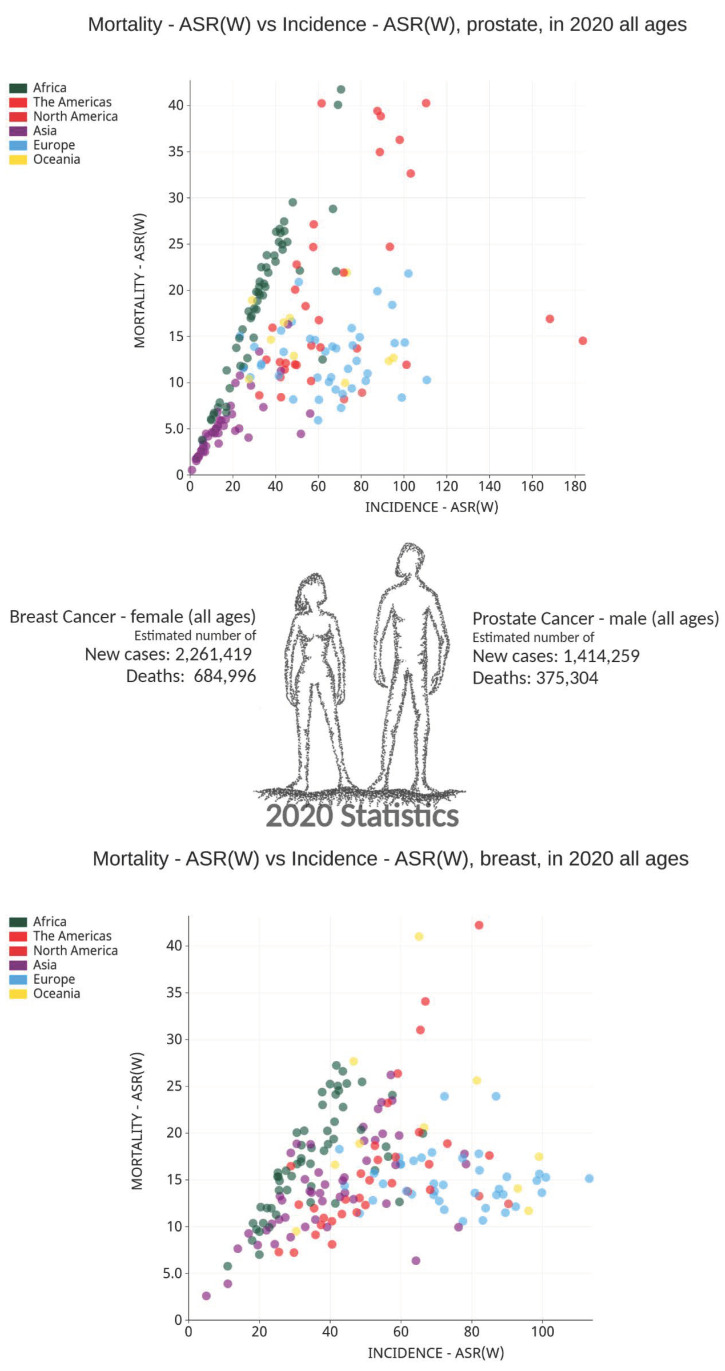
2020 Global breast and prostate cancer statistics for women and men: Estimated number of new cases and deaths for both sexes including all ages. The top graph represents prostate cancer and bottom graph represents breast cancer: age standardized rates (ASR) based estimated new cases and deaths as reported by Global Cancer Observatory (GCO). Both breast and prostate cancer are among the leading cancers contributing to both new cases and deaths in women and men. Figures taken from IARC website (https://gco.iarc.fr/today, accessed on 20 November 2021) [10,14,15].

## 3. Physiology, Complexity, and Subtypes

BC and PC consist of heterogeneous subtypes classified according to various clinical and pathological features [32,33]. The majority (~90%) of human cancers originate from epithelial tissue and are termed carcinomas [34]. In the breast, epithelial cells line the branching ductule system and associated acinar structures (i.e., terminal ductal-lobular units (TDLUs)) and, along with interlobular fat and fibrous tissue, these tissues form the mammary gland. In addition to the inner epithelial lining of secretory luminal cells, the TDLU is composed of outer contractile myoepithelial cells and basal (i.e., myoepithelial cell progenitor) and stem cells [35]. The prostate gland is also composed of divergent cell types including luminal epithelial cells that express high levels of androgen receptors and differentiation antigens (e.g., cytokeratin 8, prostate-specific antigen), basal cells that express lower levels of androgen receptors (ARs), and occasional neuroendocrine cells [36]. Mammary and prostatic glands have a double-layered glandular pseudostratified architecture, with a steroid nuclear hormone receptor (e.g., estrogen receptor and androgen receptor) expressing differentiated luminal layer and a p63 transcription factor expressing basal or myoepithelial layer [37]. Tumor protein 63 (p63, TP63 gene), a member of the p53 gene family, regulates an array of genes and signaling pathways involved in the growth and development of ectoderm derivatives including epithelial stem cells [38,39]. In a multiplex immunohistochemistry-based cancer diagnosis (e.g., determining phenotype and ruling out invasive cancer in breast and prostate tumors), p63 is a commonly used cell marker (e.g., basal cells in breast and prostatic glands) [39,40,41]. Many tumors arising from these organs are typically hormone-dependent [42] and a luminal phenotype is mainly observed in malignant cells expressing steroid hormone receptors (HR) [37]. For these hormone receptor-positive tumor subtypes, inhibition of receptor function or hormone synthesis prevents steroid hormone signaling and serves as the primary clinical intervention [37]. 

In general, the breast is composed of adipose tissue, glandular tissue, and fibrous stromal (i.e., supporting) tissue in the breast parenchyma, the superficial fascia, deep fascia, the nipple–areola complex, and skin [43,44,45] (Figure 3). It is supplied by a web of blood vessels, lymphatic vessels and nodes, and nerves [22,43,46]. A majority of the breast lymphatics drain to the axillary nodes, and lymph nodes are an important part of breast cancer diagnosis, treatment, and management [43,46,47]. The glandular tissue is exceedingly responsive to hormonal changes [22]. At the onset of puberty, estrogen and progesterone are primarily responsible for breast growth and development [25,43]. The desensitization of the hypothalamic-pituitary axis to estrogen at the onset of puberty elevates hypothalamic gonadotropin-releasing hormone (GnRH) and subsequently stimulates the anterior pituitary to release follicle-stimulating hormone (FSH) and luteinizing hormone (LH) [43,48]. Following this, estrogen and progesterone release is increased and facilitates the further development of the breast and other associated changes [25,43,49]. Insulin growth factor I (IGF1) also plays a major role in this process [49,50]. Fat to glandular tissue ratio is usually increased with age, where the postmenopausal breast has a maximal proportion [43,48,51] following from involution of the breast ductal, glandular, and stromal connective tissue after menopause [43]. 

**Figure 3 cancers-14-02954-f003:**
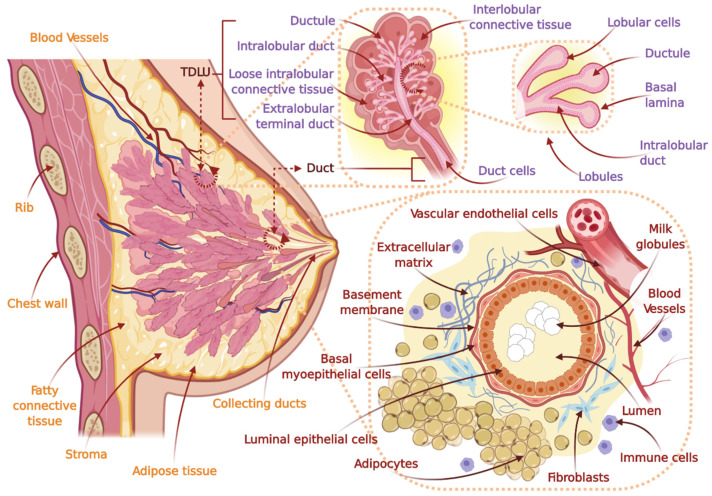
Schematic view of the human female breast and different types of interactive cells that are present within the breast tissue. The breast tissue overlays the ribs and chest muscles. The adult woman’s breast contains glandular epithelium (~10–15%) and this milk producing epithelia is contained within the surrounding adipose tissue. Multiple lobules (terminal ductules, acini, milk-producing lobules) together make up the lobes of the breast. The functional units of the breast are the terminal duct lobular units. All lobules and lobes are connected to the nipple through a branched system of ducts. Terminal ductal lobular units (TDLUs), which is a collection of ductules, intralobular duct, loose intralobular connective tissue, and extralobular terminal duct, are common sites of origin for several breast cancers. Within the stroma, two types of fibroblasts are present. Loosely connected intralobular fibroblasts surround the epithelial cells and they are subsequently encompassed by the more condensed interlobular fibroblasts. The other important cellular component of the mammary stroma is adipocytes (i.e., fat cells). The parenchymal tissue consists of epithelial and myoepithelial cells. In addition, the stromal compartment contains vascular endothelial cells and infiltrating immune cells. Stromal cells secrete factors of the extracellular matrix (e.g., collagens, hyaluronic acid, tenascins, fibronectins, proteoglycans) that are integral for the breast’s three-dimensional microstructure. Mammary ducts consist of polarized apically orientated columnar luminal epithelial cells that line (inside) ducts along with alveolar structures at the ends, as well as contractile myoepithelial cells that are orientated basally. This is enclosed by the basement membrane (BM), which forms a physical barrier that separates the epithelial and stromal compartments. BM (i.e., basal lamina) mainly consists of laminin, collagen, entactin, and proteoglycans. Myoepithelial cells that possess contractile properties and stem cells (i.e., mammary repopulating units) comprise the functionally distinct basal layer. Constituents of milk are synthesized by secretory cells that forms the alveoli, followed by secretion into the alveolar lumen. Adapted from Refs. [17,34,43,45,52,53,54,55,56,57].

Breast tumors can be cancerous (malignant) or non-cancerous (benign and usually not life-threatening) [43,45]. Non-cancerous tumors, sometimes referred to as fibrocystic changes or fibrocystic disease, may occur at any time during a woman’s lifespan [58,59]. Some women with certain benign tumors, though, may have a higher risk of developing malignant breast tumors [58]. BC predominantly originates in the ducts (ductal cancers) and less commonly the lobules (lobular cancers), which often appear in both breasts, while a small number originate in other tissues of the mammary gland [58,60,61]. Five to 15% of all invasive BCs are invasive lobular carcinomas and is the second most common BC type [62]. Inflammatory BC is an uncommon and aggressive type often associated with a warm, red, and swollen breast [60,61,63,64]. BC can spread to the other regions of the body via the lymphatic system. The presence of cancer cells in lymph nodes increases the chance of cancer cells gaining access to the bloodstream and metastasizing. However, some women develop metastases despite the absence of cancer cells in their lymph nodes [58]. Although deaths due to primary BC have declined since the 1980s due to improvements in screening methodologies, treatment options, and the availability of health care facilities, deaths have increased due to metastases and an inability to adequately treat metastatic disease with the currently available therapeutic modalities.

Weighing ~20 g and ~3 cm long, the human prostate consists of glandular tissue and requires testosterone for optimal function [65,66]. The prostate encompasses a branching duct system composed of pseudo-stratified epithelium enclosed by a fibromuscular stroma [66]. Adenocarcinomas are the most common type (95%) of PCs and originate in epithelial tissues with glandular organization on microscopic examination [28,31,65,67] (Figure 4). Sarcomas, small cell carcinomas, neuroendocrine tumors, squamous cell carcinoma, carcinoid tumors, and transitional cell carcinomas are other rare PC types [31,67]. Of all prostatic carcinomas, ductal adenocarcinoma accounts for about 3%, mixed ductal–acinar adenocarcinoma accounts for 0.2% to 0.4%, and prostatic squamous cell carcinoma (SqCCs) accounts for <0.6% [68]. Adenosquamous carcinomas are very rare and ~50% of SqCCs and adenosquamous carcinomas emerge in patients with prostatic acinar adenocarcinoma [68], which develops after androgen deprivation therapy or radiotherapy [68]. Benign prostatic hyperplasia (BPH) can occur in the transition zone or the periurethral glands of the prostate and is an age-related pathological condition [28]. Prevalence of BPH increases from 50% to 70% in men in their 50 s to 70 s [28]. Although most PC cases emerge in the peripheral zone (70%), some can arise in the transition (20%) and central zone (10%) of the prostate [28]. Adenocarcinoma typically occurs in multiple sites of the prostate [21]. High-grade prostatic intraepithelial neoplasia (“neoplastic growth of epithelial cells within preexisting benign prostatic acini or ducts” [69,70]), a premalignant lesion linked with a higher risk of coexistent cancer or slowed advancement to carcinoma, is frequently distinguished by multicentric lesions [71]. Approximately 85–90% of PCs are multifocal, which can be of monoclonal or polyclonal origin l [67,72,73]. These typically grow slowly, and controversy exists whether PC begins as a precancerous condition, such as prostatic intraepithelial neoplasia (PIN) or proliferative inflammatory atrophy (PIA), or atypical small acinar proliferation (ASAP) [31,67] (Table 1).

**Figure 4 cancers-14-02954-f004:**
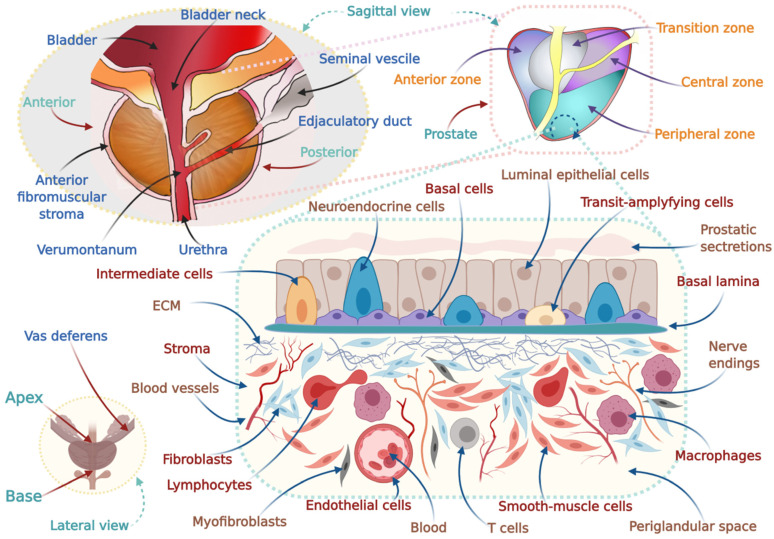
Schematic of the human prostate anatomy and cellular components within the prostate gland: The prostate, a fibromuscular glandular organ related to the male reproductive system located below the bladder with an apex, a base, a posterior, an anterior, and two lateral surfaces, is composed of 3 distinct zones having histological differences. The peripheral zone (PZ) is positioned at the posterior side constituting around 70% of the gland and is identified as the zone from where most (~75%) prostatic intraepithelial neoplasia (PIN) and carcinomas originate. Constituting 25% of the gland, the central zone (CZ) includes the ductal tube coming from the seminal vesicle to the point where it connects to the descending urethra. Cancer arising from CZ is about 5% of total prostate cancers. The transitional zone (TZ) represents roughly 5–10% of the gland, and is immediately below the bladder and encircles the transitional urethra. Approximately 20% of prostate cancers stem from TZ, which is also the region from which benign prostatic hyperplasia (BPH) develops. The normal functioning of the prostate depends mainly on the androgen, testosterone, produced by the Leydig cells within the testes and adrenal glands (i.e., androgen dependent). Dihydrotestosterone (DHT), the active metabolite of testosterone produced by prostate cell membrane 5α-reductase isoenzymes, binds to androgen receptors (i.e., adrenoceptors) to conduct signaling pathways that involves tissue and organ regulatory effects. Several cell types including basal, luminal, and neuroendocrine cells, and two in-between phenotypes (transit-amplifying cells, which are more basal-like phenotype non-secretory cells, and intermediate cells, which are more luminal-like phenotype secretory cells), are the main components of the epithelial compartment. Epithelial tissue is backed by a stroma consisting of extracellular matrix (ECM), blood vessels, immune system cells, nerve fibers, and stromal cells such as fibroblasts, myofibroblasts, and smooth muscle cells (most abundant). The stroma or periglandular space acts as the boundary of the gland. High levels of androgen receptors (AR) are expressed in the secretory columnar shaped cells called luminal cells, and can be considered the major prostate or epithelial cell type because it forms the exocrine compartment that is involved in the secretion of prostate-specific antigen (PSA) and prostatic acid phosphatase (PAP) into the acinar lumen. The non-secretory cuboidal shaped cells having low or undetectable AR located along the basement membrane are known as basal cells. Neuroendocrine cells are non-secretory, differentiated, androgen-insensitive cells that express CD56, chromogranin A (CgA), synaptophysin, neuron specific enolase (NSE) and neuropeptides (e.g., bombesin, calcitonin, and neurotensin) and are rare (~1% of the epithelium). The apex is aimed downward and connects with the superior fascia of the urogenital diaphragm. The base is aimed upward close to the bladder’s inferior surface, and it is partly continuous with the bladder wall. Adopted from Refs. [66,74,75,76,77,78,79,80,81,82,83].

**Table 1 cancers-14-02954-t001:** General classification of prostatic carcinoma. Adapted from Refs. [68,84].

The Main Type of Neoplasm	Subtype (s)
Glandular neoplasms	Acinar adenocarcinoma, Ductal adenocarcinoma, Intraductal carcinoma
Basal cell carcinoma	-
Urothelial carcinoma	-
Squamous neoplasms	Squamous cell carcinoma, Adenosquamous carcinoma
Neuroendocrine neoplasms	Small cell neuroendocrine carcinoma, large cell neuroendocrine carcinoma, Adenocarcinoma with neuroendocrine differentiation

Molecular taxonomy is continually changing and progressing [85]. The best example is the different molecular subtypes of breast carcinomas that have emerged during the past decade. Apical (luminal) and basal orientations are major characteristics of epithelial tissues, and tumors with epithelial tissue origin (e.g., carcinomas) can display varying ratios of luminal and/or basal differentiation owing to this dichotomy of epithelial tissues [33]. Therefore, the overall prognosis and treatment response of a certain tumor can be influenced by the basalness and/or luminalness of the malignant tissue [33]. This makes the understanding and clarification of this indispensable biological difference very important [86]. The literature reveals clinically relevant basal and luminal subtypes of several dissimilar carcinomas [33].

Using data from gene expression profiling, four intrinsic BC subtypes with clinical implications were initially identified by Perou and colleagues [87]. Subsequently, the ‘Prediction Analysis of Microarray 50′ (PAM50) classification was introduced by Parker and colleagues in 2009 [88]. Researchers have documented evidence for the inclusion of 5 molecular/intrinsic subtypes based on microarray and hierarchical clustering analysis (i.e., genomics-based molecular classifications [89,90]). These are triple-negative/basal-like (HR- and HER2-), HER2 enriched (estrogen-receptor [ER] and progesterone-receptor [PR] negative), luminal A (ER and/or PR positive, HER2 negative, low protein Ki-67), luminal B (ER and/or PR positive, and either HER2 positive or HER2 negative, high protein Ki-67), and normal-like (similar to luminal A, but with a poorer prognosis than luminal A) [89,91]. In addition, HR+/HER2+, ER+/HER2− basal (intermediate between the ER+/HER2− luminal and ER− basal subtypes), and ER+/HER2− luminal subtypes have been reported [90,92]. The commonly used marker to evaluate proliferative index in cancer is Ki67, with no defined cutoff values for “high ki67 index” [93]. 

Highly differentiated luminal secretory cells and a small fraction of basal cells can be found in the prostate gland epithelium and, therefore, adenocarcinomas may arise from both luminal and basal tumor progenitor cells [94]. Although the origins of human PC cell type is controversial, frequently diagnosed prostate tumors tend to have a luminal phenotype [94]. Analogous to BC, luminal and basal subtypes of PC have been explained using a slightly changed PAM50 algorithm [33]. Zhao and colleagues established the PC basal/luminal classification and indicated that the androgen receptor (AR) pathway was enhanced in luminal (A and B) tumors compared to basal tumors, using a gene set enrichment analysis [88,95]. The luminal B subtype has the highest PAM50 proliferation score and poor clinical and pathologic characteristics, where clinical endpoints such as distant metastasis-free survival, biochemical recurrence-free survival, PC specific survival, and overall survival had consistently worse outcomes [88]. Interestingly, this is the opposite compared with BC where basal like type is associated with a poor prognosis [88]. Compared to other cancers, PC has a relatively lower mutation rate including few chromosomal losses and gains. Therefore, a wide variety of studies suggest two prominent molecular groups of PC that exist [36], which are characterized by (a) presence of E-26 transformation-specific (ETS) related gene (ERG) rearrangements and features of chemoplexy, and (b) absence of ERG rearrangements and deletion of CDH1 and/or mutation frequency of E3 ubiquitin ligase adapter speckle-type pox virus and zinc finger protein (POZ protein) (i.e., SPOP) [36,96,97,98]. 

PC is considered a highly heterogeneous disease because some tumors cannot be categorized purely based on a molecular subtype [83,95,99]. A recent study on epithelial tumors reported the use of a modified PAM50 clinical-grade classifier to subtype 8764 tumors covering 22 distinct carcinomas into luminal A, luminal B, and basal-like tumors identifying global similarities in the genome, transcriptome, drug sensitivity, and clinical outcomes, and highlighting the biological and translational significance of luminal versus basal subtypes (i.e., pan-carcinoma luminal/basal subtyping) across carcinomas [33]. Neuroendocrine carcinoma, a very distinct entity of the breast (i.e., breast carcinoma with neuroendocrine differentiation or neuroendocrine breast carcinoma) and prostate, is also a rare possibility with very little data from pre-clinical and clinical trials on its better management [100,101,102]. These neoplasms represent a diversified collection of tumors that may persist in association with adenocarcinomas or as pure carcinomas and may not always correlate with advanced disease [101]. Interestingly, patients who have had pancreatic neuroendocrine cancer are at an increased risk for prostate and breast cancer [103]. Therefore, the complex relationship between BC or PC and neuroendocrine cancer has been of special recent interest [100,101].

## 4. Pathology, Risk Factors, and Treatment

Disease burden is the salient determinant of survival [104]. Determining the cancer stage and codifying using a classification system is fundamental to the prognosis of PC and BC [104]. The TNM staging system (tumor-node-metastasis) is widely employed for most solid tumor progression where each letter represents various stages (T- local growth (4 stages), N- lymph node status (3 stages), and M- distant metastasis (2 stages)), for a total of 24 different TNM combinations [104,105,106,107,108]. Separating tumors into different categories with varying behavior and prognosis based on morphology was the basis of the traditional classification system [85]. However, novel procedures based on molecular methods have evolved to improve tumor classification [85]. In general, ‘histological grade’ based on tumor differentiation (tumor vs normal tissue degree of similarity) includes Grade X (unknown/undetermined grade), Grade 1 (highly differentiated), Grade 2 (medium-differentiated), Grade 3 (low-differentiated), Grade 4 (undifferentiated), plus other identifiers including lymphatic invasion (L), venous invasion (V), and residual tumor (R) [105,107]. The reader is directed towards other reviews [104,106,107,109] for further information on staging and classification of cancer. 

The development of BC is associated with several risk factors. (Figure 5) These include a personal history of invasive BC (either ductal carcinoma in situ (DCIS) or lobular carcinoma in situ (LCIS), or benign tumors or cysts) [60,61,110], a family history of first-degree relatives with BC, inherited changes in breast cancer gene 1 and 2 (BRCA1 and BRCA2), dense breast tissue, early menstruation, late onset menopause, use of hormonal treatment for menopausal symptoms, nonparity or having a first child later in life, overexposure to ionizing radiation (e.g., radiation therapy to the breast or chest area), excessive alcohol consumption, obesity, lack of exercise, ethnicity, and age [60,61,110,111]. Approximately 5–10% of all BCs are classified as hereditary and certain mutated genes are more common in specific ethnic groups [21,25,112]. Cancer is a disease that can lead to financial toxicity [111,113,114]. The risks associated with developing BC include changeable life style factors such as alcohol consumption, dietary practices, obesity, and physical inactivity as well as unchangeable factors such as age, race, gender, and family history [115] and can contribute to the economic burden [111,114,116,117]. Considering its complexity, the histological subtypes of BC include invasive ductal carcinoma (i.e., ‘no special type’) and lobular carcinoma, as well as preinvasive ductal carcinoma in situ and lobular carcinoma in situ (i.e., lobular neoplasia) [25]. It is estimated that the worldwide healthcare expenditure on BC attributed to physical inactivity was (USD) 2.7 billion [111,118,119].

An initial diagnosis of BC typically follows from a physical exam and history, clinical breast exam, mammogram, ultrasound, magnetic resonance imaging, blood chemistry, and biopsy [21,60,61,112]. Once a cancer is suspected or found, further tests such as estrogen-progesterone receptor test, human epidermal growth factor type 2 receptor (HER2/neu) test, and multigene tests are performed to classify BC as hormone receptor-positive or negative (estrogen (ER) and/or progesterone receptor (PR)), HER2/neu receptor-positive/negative, or triple negative (ER, PR, and HER2/neu negative) [60]. Disease stage, type, ER-PR-HER2/neu levels within the cancerous tissues, tumor growth rate, reoccurrence rate, possibility of relapse, woman’s age, overall health, and menopausal status, and the point and time of diagnosis all contribute to the prognosis and treatment of BC [60,61]. 

Diagnosis, prognosis, and treatment are complicated by the tremendous clinical and genetic heterogeneity of BC [120]. Aneuploidy is reported to be a hallmark of BC [121]. Alterations in copy numbers are often observed at pre-malignant stages because 15 to 44% of atypical ductal hyperplasia are aneuploid [122]. The frequently mutated and/or amplified genes related to BC include phosphatidylinositol-4,5-bisphosphate 3-kinase catalytic subunit alpha (PIK3CA), transcription factor tumor protein p53 (TP53), phosphatase and tensin homolog (PTEN), proto-oncogene myelocytomatosis (MYC), cyclin D1 (CCND1), human epidermal growth factor receptor 2/proto-oncogene C-ErbB-2/Erb-B2 receptor tyrosine kinase 2 (ERBB2), fibroblast growth factor receptor 1 (FGFR1), and transcription factor GATA binding protein 3 (GATA3) [25,123,124,125,126,127,128,129,130,131,132,133]. To address the heterogeneity, classification systems have evolved to improve treatment choices and prognosis [134]. Currently, BC is classified according to its histology based on cellular and tissue architecture and growth pattern, its intrinsic molecular basis, which is usually based on microarray analysis, and its functional basis often based on the tumor-initiating cells. To understand the local and metastatic status of BC, further tests are required including x-ray, computerized tomography (CAT) scan, bone scan, positron emission tomography scan, and sentinel lymph node biopsy [60,61]. Depending upon the extent of invasiveness, BC is classified as Stage 0 (carcinoma in situ), Stage I, Stage II, Stage IIIA, Stage IIIB, Stage IIIC, or Stage IV (the metastatic stage) [58]. The 5-year average survival rate is 90% for women with the invasive type, and the average 10-year survival rate is 83% [135]. For localized cancer within the breast, the 5-year survival rate is 99% with 62% of the overall cases being diagnosed at this stage [135]. Patient survival rate is heavily affected by disease progression, and beyond stage IIIB the 5-year survival rate falls below 50% [21,58]. This current model for breast neoplasm classification will likely undergo further revision as new discoveries are made.

Epithelial cells, smooth muscle cells, autonomic nerve fibers and associated ganglia, endothelial cells, immune cells, and fibroblasts can shape the biology and clinical behavior of the prostate and associated malignancies [136]. Based on histological patterns of prostate adenocarcinoma, Gleason and Mellinger in 1974 first introduced the ‘Gleason grading system’, which is still commonly used to define PC aggressiveness, although the original grading system has undergone refinement over the years [65,68,136]. The new grading system is given a ‘grade group’ from 1 to 5 with assigned Gleason scores ranging from 1 to 10 [65,109]. For 1 through 5 grade groups, 96%, 88%, 63%, 48%, and 26% are the respective 5-year biochemical risk-free survival rates [68]. Additional histological features include an increase in mitotic figures, abnormally enlarged cell nuclei with large nucleoli, glandular infiltrative growth pattern, lack of a basal cell layer, intraluminal crystalloids, amphophilic cytoplasm, amorphous pink secretions, and intraluminal wispy blue mucin [65,137]. Reasonable clinical outcomes are observed in patients with localized disease. However, metastatic PC tends to have a poor prognosis leading towards a 5-year survival rate of 30% [138]. A considerable increase in genome-wide copy number alterations is observed, but only a modest increase in mutations is detected between prostate primary tumors and metastatic castration-resistant PC [136]. Pathways associated with DNA repair, AR, PI3K–PTEN, WNT (i.e., Wingless-related integration site), and the cell cycle are the main targets of the customary genetic alterations in almost all metastatic PCs and a significant portion of primary tumors [136]. Heterogeneity on the functional level is not uncommon in PC, particularly relating to the differentiation status and lineage plasticity [136]. 

Several standard treatment options are currently available for the treatment of BC including surgery, radiation therapy, chemotherapy [58], hormone therapy [58], and a relatively newer approach of targeted therapy [58,60,61] (Table 2). High-dose chemotherapy with stem cell transplant is another experimental treatment option in clinical trials [139,140,141]. Often patients experience different treatments simultaneously or sequentially during the duration of their treatment plan. One major drawback of traditional treatments is their varying and dangerous side effects such as lung inflammation, lymphedema, heart failure, blood clots, dental issues, bone loss and osteoporosis, cataracts, musculoskeletal symptoms, sexual difficulties and infertility, menopausal symptoms, absence of menstrual periods, headaches, memory loss and cognitive function, fatigue, peripheral neuropathy, and risk of other cancer types, which may manifest during treatment or months or years post treatment [60,61,142].

Early PC is typically asymptomatic but may show similar symptoms as BPH, such as nocturia, hematuria, dysuria, and problems with urination [65]. The development of PC is associated with a number of risk factors. (Figure 5) These include increased age, ethnicity, and race (more common amongst African descent men, and low occurrence among Asian populations), geography (more common in North America, North, and Western Europe, Australia, Caribbean, Southern Africa, and South America), family history of PC, and genetic changes (mutations especially in BRCA2 or men with hereditary non-polyposis colorectal cancer) [27,31]. Genetic predisposition might play a role in differences amongst the ethnic groups. For example, chromosome 8q24 variants that are associated with elevated PC risk are more common in African American men [143]. Additionally, variations in genes that control carcinogenesis (e.g., ephrin type-B receptor 2 (EphB2), and B-cell lymphoma 2 (BCL2)) may be responsible for aggressive PC forms in men with African descent [143]. Differences in incidence and mortality rates among men of various ancestry, as well as the dissimilarities between men of the same ethnicity and/or race living in separate countries, hints at the involvement of genetic and environmental factors [144]. Family history associates with about 20% of PC cases [143]. However, shared genes and exposure to a similar pattern of specific environmental carcinogens and common lifestyle habits can be included in this group [143]. 

Other factors with association to cancer risk include diet (increased intake of dairy products, saturated animal fat and red meat, less consumption of vegetables and fruits, vitamin deficiencies, and coffee consumption), smoking, vasectomy, inflammation, sexually transmitted diseases, infections, chemical, and ionizing radiation exposures, alcohol abuse (>15 g ethanol/day), lower ejaculatory frequency, physical inactivity, and obesity [31,115,143,145]. Studies on immigration from developing countries (low risk) to developed countries (high risk) suggest that change to the western lifestyle can increase cancer incidence [143]. Apart from androgens, IGF-1 is also linked to an increased PC risk, while use of aspirin and the 5-α reductase inhibitors (treatment of BPH) may decrease risk [31]. 

Depending upon the PC case, treatment option(s) may include active surveillance, surgery, radiation therapy, chemotherapy, cryotherapy (cryosurgery), hormone therapy, vaccines, bone-directed treatments, and pain medication [31] (Table 2). Initial PC progression is aided by the androgen hormone, which stimulates proliferation and/or restricts apoptosis of PC cells [146,147]. Consequently, hormone deprivation through surgery or castration is considered a gold standard for PC treatment with success in about 70% of patients diagnosed with primary PC [148]. Unfortunately, androgen deprivation is not effective in patients with advanced PC. In advanced disease, androgen-receptor signaling is reactivated over time with notable increases in serum prostate specific antigen (PSA) concentration [149]. Prostate tumors that maintain high proliferation rates in the absence of androgen are identified as castration-resistant (CR) and are generally resistant to most chemotherapeutic drugs and show poor prognosis [149]. CRPC is usually a late-stage phenomenon of PC, and is unresponsive to androgen deprivation therapy (ADT) [147]. Poor prognosis with an average survival time of 16–18 months makes CRPC a challenging issue [150]. Resistance is due to enhanced expression of ARs, which leads to over-activation of the receptor in the presence of low androgen levels [151]. Enhanced AR copy number is reported in 25–30% of patients with CRPC [152]. As well, mutations of the AR gene increase the number of ligands which can activate the receptor to enhance PC cell proliferation [147,153]. Interestingly, increased AR levels can convert AR antagonists to agonists [154]. AR-regulated proteins, such as the PSA, increase in CRPC despite low levels of serum testosterone [155]. Therefore, development of PC can be classified into 3 stages: (i) precancerous state, intraepithelial neoplasia, (ii) androgen-dependent adenocarcinoma with two stages, latent and clinical, and (iii) androgen-independent adenocarcinoma or castration-resistant [36]. Furthermore, de novo steroidogenesis during progression results in high androgen levels within the tumor to further aid resistance to ADT, and enhanced expression of the enzymes involved in the conversion of cholesterol to dihydrotestosterone has also been reported [156]. Men with CRPC that failed first line treatment usually display cross-resistance to a wide variety of drugs [147]. About 15–20% of advanced treatment-resistant PCs display a loss of dependence on AR signaling [157]. This may become evident clinically, through a transformation of adenocarcinoma to a castration-resistant neuroendocrine PC (CRPC-NE), along with several detectable genetic (e.g., TP53, retinoblastoma transcriptional corepressor 1 (RB1), cylindromatosis deubiquitinase lysine 63 (CYLD), AR) and epigenetic modifications (e.g., hypo- and hypermethylation) [157]. The variability between PC incidence and mortality is noteworthy because autopsies reveal around 60–70% elderly men experience histological PC, though most are silent and have no clinical progression [158,159].

**Table 2 cancers-14-02954-t002:** Currently trending [165,166,167] intervention options for breast and prostate cancerous and non-cancerous conditions. (Adapted from Refs. ‘cancer.ca’, ‘cancer.org’, and others [61,168,169,170,171,172,173,174,175,176,177,178,179,180,181,182,183]).

Type	Therapy Type	Situation/Condition Used	Example(s)
Breast	Surgery	Depends on: size of the breast, size, and location of the tumor, spread to lymph nodes (e.g., sentinel lymph node biopsy or axillary lymph node dissection), prior treatments, breast reconstruction, a need to relieve symptoms of advanced disease	Breast-conserving surgery (lumpectomy, quadrantectomy, partial mastectomy, segmental mastectomy), mastectomy, lymph node surgeries, other (e.g., oophorectomy, deep inferior epigastric perforator (DIEP) flaps, noninvasive tissue oximetry, muscle-sparing (MS) free transverse rectus abdominis musculocutaneous (TRAM) flap, robot-assisted surgery)
Breast	Radiation therapy	Often after breast-conserving surgery, sometimes after mastectomy, treat metastases (e.g., bones, lungs, or brain), in addition to other treatments	External beam radiation therapy (EBRT), whole breast and tumor bed radiation, accelerated partial breast irradiation (APBI) like intraoperative radiation therapy (IORT), 3D-conformal radiotherapy (3D-CRT), intensity-modulated radiotherapy (IMRT), and brachytherapy (internal radiation—intracavitary brachytherapy, interstitial brachytherapy), lymph node radiation, chest wall radiation, hypo-fractionated radiation therapy
Breast	Chemotherapy	After surgery (i.e., adjuvant chemotherapy), prior to surgery (i.e., neoadjuvant chemotherapy), prevent recurrence, advanced (e.g., locally advanced cancers, residual disease) and/or metastatic disease, combination therapy, treatment response	Dose-dense chemotherapy, 5-fluorouracil, cyclophosphamide, platinum agents such as cisplatin and carboplatin, taxanes (e.g., paclitaxel, docetaxel, albumin-bound paclitaxel), anthracyclines (e.g., doxorubicin, epirubicin, pegylated liposomal doxorubicin), vinorelbine, capecitabine, gemcitabine, mitoxantrone, ixabepilone, eribulin
Breast	Hormone therapy/endocrine therapy	Treat HR + cancer, lower estrogen levels, often after surgery, as adjuvant therapy, as neoadjuvant therapy, treat metastatic disease in postmenopausal women, treat advanced cancer with no prior treatment with other hormone therapy or upon response failure of other hormone drugs, for ovarian ablation, for combination therapy (e.g., LHRH agonist, CDK 4/6 inhibitor, PI3K inhibitor)	Selective estrogen receptor modulators (SERM) such as tamoxifen and toremifene, anti-estrogen agents or selective estrogen receptor degrader (SERD) like fulvestrant, aromatase inhibitors that interfere with estrogen production (e.g., letrozole, anastrozole, exemestane), luteinizing-hormone releasing hormone (LHRH) agonists (e.g., goserelin, leuprolide), other drugs (progesterone-like drug „megestrol acetate “, high dose of estrogen, androgens)
Breast	Targeted therapy, Immunotherapy, & Other	Combination therapy, treat early-stage (aka neoadjuvant chemo) and/or advanced cancer	Monoclonal antibodies (e.g., trastuzumab and pertuzumab), antibody conjugates (e.g., Ado-trastuzumab emtansine, Fam-trastuzumab deruxtecan), PD-L1 immunotherapy (e.g., atezolizumab), HER2 kinase inhibitors (lapatinib, neratinib) mammalian target of rapamycin (mTOR) inhibitors (e.g., everolimus), cyclin-dependent kinase/CDK4-6 inhibitors (e.g., abemaciclib, palbociclib ribociclib), PI3K inhibitors (e.g., alpelisib), PARP inhibitors (e.g., olaparib, talazoparib), regenerative medicine (e.g., stem cell associated therapy/cytotherapy, gene therapy, tissue engineering)
Prostate	Active surveillance/observation	Disease risk, disease progression, every 3 to 6 months, based on Gleason score, low PSA level (<10 ng/mL), location and size of the lesion	Watch for any signs and symptoms, routine blood work, prostate-specific antigen (PSA) test, digital rectal exam (DRE), imaging tests (e.g., CT scan, MRI, bone scan)
Prostate	Surgery	Lymph node status, PSA level, prostate biopsy results, other factors, treat non-cancerous enlargement of the prostate (i.e., BPH)	Open radical prostatectomy (radical retropubic prostatectomy, radical perineal prostatectomy), laparoscopic prostatectomy (laparoscopic radical prostatectomy, Robotic-assisted laparoscopic radical prostatectomy), Transurethral resection of the prostate (TURP), pelvic lymph node dissection (PLND)/pelvic lymphadenectomy, nerve-sparing radical prostatectomy, orchiectomy
Prostate	Cryotherapy/cryosurgery	Recurrent cancer, radiation therapy	Cryoablation (freeze and kill cancer cells or most of the prostate)
Prostate	Radiation therapy	Stage of the disease, other factors, metastases to bone	External beam radiation therapy (EBRT), brachytherapy (internal radiation/seed implantation/interstitial radiation therapy), permanent (low dose rate) brachytherapy, temporary (high dose rate) brachytherapy, three-dimensional conformal radiation therapy (3D-CRT), intensity modulated radiation therapy (IMRT), image guided radiation therapy (IGRT), volumetric modulated arc therapy (VMAT), stereotactic body radiation therapy (SBRT), proton beam radiation therapy, systemic radiation therapy, radiopharmaceuticals (strontium-89, samarium-153, radium-223, iodine-125, palladium-103, iridium-192, cesium-137)
Prostate	Chemotherapy	Metastases, failure of hormone therapy, castration-resistance, combination therapy, palliative chemotherapy	Docetaxel, cabazitaxel, paclitaxel, vinorelbine, doxorubicin, epirubicin, etoposide, mitoxantrone, estramustine, cisplatin, carboplatin
Prostate	Hormone therapy/androgen suppression therapy	Before radiation, initial treatment, metastases, advanced disease, cannot be cured by surgery or radiation, cancer remains or comes back after surgery/radiation, high Gleason score, castration-resistance, high PSA level, combination treatment, combined androgen blockade (CAB), triple androgen blockade (TAB), for orchiectomy	Androgen deprivation therapy (ADT), orchiectomy (i.e., surgical castration), medical castration (i.e., LHRH agonists/LHRH analogs/GnRH agonists) (e.g., leuprolide, triptorelin, goserelin, histrelin), LHRH antagonists (e.g., Degarelix), CYP17i inhibitor (e.g., abiraterone), anti-androgens (i.e., androgen receptor antagonists) (e.g., flutamide, bicalutamide, nilutamide, enzalutamide, apalutamide, darolutamide), other androgen suppressing drugs (e.g., estrogens, ketoconazole), 5-alpha reductase inhibitors (e.g., finasteride, dutasteride)
Prostate	Immunotherapy	Advanced prostate cancer, no response to hormone therapy, lynch syndrome	Vaccines (e.g., sipuleucel-T), immune checkpoint inhibitors such as PD-1j inhibitor (e.g., pembrolizumab)
Prostate	High intensity focused ultrasound (HIFU)	Recurrent cancer, after radiotherapy	Experimental treatment
Prostate	Other	Stage of the disease, other factors, metastases to bone, relieve pain	Bisphosphonates (e.g., zoledronic acid), antibodies (e.g., denosumab), pain medication, steroid drugs (aka corticosteroids) (e.g., prednisone, dexamethasone), regenerative medicine

Abbreviations: Hormone receptor positive (HR+); Phosphoinositide 3-kinase (PI3K); Human epidermal growth factor receptor 2 (HER2); Poly ADP-ribose polymerase (PARP); Computed tomography (CT); Magnetic resonance imaging (MRI); Benign prostate hyperplasia (BPH); Member of the cytochrome P450 superfamily involved in metabolism and steroid synthesis (CYP17), Programmed cell death protein-1 found on immune system T cells (PD-1).

## 5. Future Landscape

Cancer presents a substantial and increasing impact on many populations and health care systems around the world. The number of patients with cancer will continue to rise over the next several decades, mainly due to the influence of changing population demographics, like aging and growth [184,185]. During the recent past, cancer-related mortality rates have decreased, and survival has improved mostly owing to improvements in diagnoses and advancements in treatment modalities [185,186]. In general, early diagnosis plays a key role in managing cancer. However, early diagnosis remains an important hurdle to effectively address the burden of cancer in patients. Early detection is not always successful as the processes involved in tumorigenesis commence prior to the manifestation of any recognizable pre-malignant morphological changes as well as before reaching the stage of a clinically detectable cancer or lesion [187]. Additionally, under certain instances, some cells that harbor numerous genetic and epigenetic changes do not progress into malignancy right away, have delayed progress, or do not progress at all [122,188]. This makes the processors of the development of cancer or metastatic disease unpredictable and difficult to detect clinically. 

The past couple of decades has witnessed rising incidence rates [186] and notable variations in survival rates across different geographical areas and types of cancer. Effective strategies for managing breast and prostate malignant disease remain elusive. Hence, revisiting challenges and considerations responsible for cancer control is relevant in advancing the field of disease management. Treatment has evolved with the sequential progress of four elementally recognized areas of interventions including surgery, radiotherapy, chemotherapy, and precision therapies (i.e., targeted cancer therapies) [186,189,190]. The latest addition to this list (fifth type of intervention), is immuno-oncology treatment [186]. Currently, these intervention modes are commonly and routinely used in combination to ensure the destruction and removal of malignant tissues and cells from the patient’s body [186].

At present, failure of treatment contributes considerably to cancer-related morbidity and mortality [27]. Although our enhanced understanding of somatic genetic alterations or “driver” mutations in oncogenic signaling cascades have given rise to a variety of clinical and experimental therapies [191], patient response to anti-cancer treatment is inconsistent with frequently observed dissociated or mixed responses, where some tumors/cancer cells positively respond to treatment whereas others do not [192,193]. Disease management in patients displaying such dissociated responses is more complex because interruptions in treatment modalities can lead to the growth of non-responding, and/or responding metastases, thus leading healthcare practitioners to continue treatment past disease progression [192,194]. As well, an inability to remove all cancer cells may unintentionally give rise to disease resistant cells, where selective pressures on cancer cell survival by these treatments results in their expansion [195]. Undetectable cancer cells that persist in the patient after treatment (i.e., existence of minimal residual disease) can eventually and unpredictably give rise to relapses and metastases [192,196]. Such drug resistance (either intrinsic or acquired) prevents anti-cancer therapies from executing stable and thorough responses to the therapeutic intervention and, rather, confers a proliferative and/or survival advantage [196,197,198]. 

Multidrug resistance (MDR) is reported to cause over 90% of deaths among cancer patients undergoing or having undergone broad spectrum chemotherapeutic therapy or treatment with other newer therapeutics such as targeted drugs [199,200]. MDR mechanisms include changes in drug metabolism or cellular efflux, growth factors, genetic factors, epigenetic alterations, cellular plasticity, and increased capacity of DNA repair [193,199,200]. There can also exist some minor population(s) of malignant cells that evade cell death mechanisms from various anti-cancer therapies by plunging into a reversible slow proliferation state (or slowly cycling quiescent cells). These are generally identified as cells in a drug tolerant persister (DTP) state [191,201]. The literature surrounding the nature of the proliferation status of persister cells is unclear [191,200]. This DTP state seems to give the ability of certain malignant cells without bona fide resistance mechanisms to survive drug treatment long enough to develop additional mechanisms of acquired drug resistance [191,192,201]. Interestingly, such persister cells do not seem to harbor classical drug resistance driver alterations [191,192,202], but this partial resistance phenotype seems to be transient and reversible following drug removal [191]. 

Breast and prostate cancer are known to be associated with DTP [200]. DTP cancer cells enact global transcriptional reprogramming by utilizing nongenetic mechanism(s) such as distinct metabolic patterns, maintenance of stemness, transient cell cycle arrest, and orchestration of redox signaling [192,196,198,202,203,204]. It is important to note that DTP quiescent cell status is different from cancer cell dormancy status, which is generally characterized as neoplastic cells that may or may not be genetically altered and that remain quiescent for extended time periods independent of the presence of any drugs [192,193]. Adaptable cell metabolism, modified cell proliferation, plasticity, and modulation of the microenvironment, are four non-mutually exclusive and often co-existing strategies that DTP cells exploit to survive [192,200,205,206]. Recent studies reveal that DTP cells can be sensitive to ferroptosis (excessive accumulation of lipid peroxide) owing to their propensity to accumulate highly polyunsaturated lipids and phospholipids. As well, DTP cells are sensitive to the unusual cytoplasmic vacuolization process of methuosis (or excessive macropinocytosis) and disequilibrating Niemann–Pick C1-like 1 (NPC1L1)-mediated redox control by promoting the permeation of anti-cancer agents and inciting catastrophic and expeditious fluid uptake [192,196,197,198,207,208]. 

The presence of drug resistant and DTP cell populations speak to a need to exploit multiple approaches to inhibit proliferation and promote cancer cell death. The literature identifies many cell death or anti-survival modalities, and these include accidental cell death, anoikis, autophagy-dependent cell death, autosis, efferocytosis, entotic cell death, ferroptosis, immunogenic cell death, lysosome-dependent cell death, mitochondrial permeability transition (MPT)-driven necrosis, necroptosis, NETotic cell death, parthanatos, pyroptosis, mitotic catastrophe and mitotic death, extrinsic apoptosis, intrinsic apoptosis, and methuosis [209,210,211,212,213,214,215,216,217,218]. With the availability of a vast array of clinical and experimental synthetic and natural compounds and drug delivery modes, their rational combination has the potential to target a multitude of cell death mechanisms [207,208,209,210,211,212,213,214,215,216,217,218,219,220,221,222,223,224,225,226,227,228,229,230,231] to bring about the death of all cancer cells including cancer stem cells, dormant cells, circulating cancer cells, micro-metastases, and DTP and muti-drug resistant malignant cells (Figure 6).

Our incomplete understanding of the underlying molecular underpinnings of cancer and inability to effectively use consistent disease taxonomy classification systems [232,233] underpin the slow advancements that have been made in effectively treating cancer. It is evident that pharmacological interventions against cancer continuously change as our view of cancer shifts with further advancements in our knowledge of the molecular and cellular characteristics of cancer and the establishment of the cancer hallmarks [234]. The cancer-specific trends observed globally signal an urgent need for effective therapeutics, but also suggests an opportunity to harness the potential of cancer prevention. Until such therapies and prevention strategies are available, though, individualization of therapy, improvements in diagnosis, and use of combination therapies will be indispensable in helping to reverse the growing trend in cancer incidence, morbidity, and mortality. 

**Figure 6 cancers-14-02954-f006:**
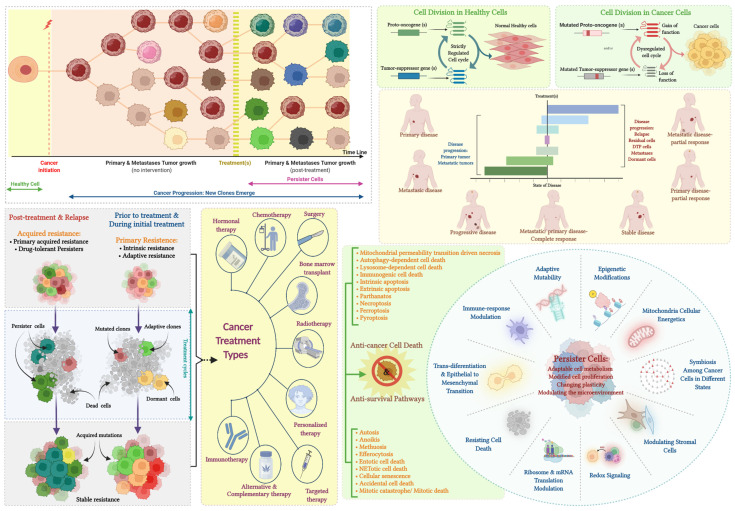
Resistance to cancer therapies, clonal selection, and cancer cell survival: Cancer treatment can involve either a single therapy or combination of surgery, targeted therapies, radiotherapy, broad spectrum chemotherapeutics, immune-therapeutics, hormonal therapy, personalized therapy, bone marrow transplants, and complementary and alternative medicine. Malignant cells display resistance to treatments through a myriad of genetic and non-genetic mechanisms. The loss of function of tumor-suppressor genes and the gain of function of proto-oncogenes provides a survival advantage to cancer cells. Various selective pressures from scarcity for nutrients, and oxygen, treatment modalities, patient lifestyle factors, and body tissue environments can help certain cancer cells gain features (i.e., clonal selection) that support survival and advance disease progression. The ability to adapt allows cancer cells ability to survive during any stage of the disease progression. Therapeutic resistance may occur at the time of initial therapy (i.e., primary resistance) or post therapy (i.e., acquired resistance). Primary resistance may result from intrinsic and/or adaptive resistance owing to ineffective targeting of the oncogenic drivers and/or rapid rewiring of oncogenic signaling after the initial suppression or may be due to non-therapy related selective pressures. Due to heterogeneity within a tumor mass, tumors can harbor rare subclones with treatment resistance mechanisms even before the initiation of therapy. Alternatively, in acquired drug resistance, after initial treatment response, relapse of the disease might occur through clonal selection. Resistant cells that exist prior to treatment may expand due to treatment mediated selective pressures and eventually evolve further and acquire further mutations. Drug-tolerant persister (DTP) cells that acquire resistance mechanisms (without de novo genetic mutations) during therapy are a major stumbling block in achieving successful treatment. Such residual persistent cells are capable of adapting to their micro-environment where they can stay hidden for extended periods of time and in due course can act as a reservoir for the instigation of genetic resistance. The presence of DTP cells can vary across different types of therapeutic responses and a patient may possess more than one type of DTP cell within a single tumor and/or multiple metastases. At the macroscopic level, a patient may show complete response (tumor size reduced 100%) or partial response (≥30% metastases size reduction) or stable disease (sum of metastases size between −30% and +20%) or progressive disease (increased tumor size ≥ 20%). DTP cells are known to have characteristics such as epigenetic modifications, mitochondrial cellular energy modulation, symbiotic relationships with other malignant cells for survival benefit, modulation of surrounding tissue stromal cells, control of REDOX signaling and reactive oxygen species, influencing ribosomes and protein translation, resistance to cell death mechanisms, trans- differentiation and epithelial to mesenchymal transition capability, ability to modulate immune responses, and the ability to further mutate. Given these characteristics, we can categorize the ways in which persistent cancer cells can evade treatment into four often co-existing and non-mutually exclusive strategies: (a) adaptable cell metabolism, (b) modified cell proliferation, (c) changing cellular plasticity, and (d) modulating the microenvironment. Molecular mechanisms underlying distinct regulated cell death pathways show remarkable interconnectivity. This implies that targeting a single cell death pathway maybe ineffective in eliminating malignant cells and, therefore, activation of multiple cell death mechanisms and/or anti-cell survival mechanisms to target cancer cells may bring about improved anti-cancer responses, increased patient survival, and greater clinical success. Adopted from Refs. [191,192,197,198,199,201,202,204,209,213,217].

## 6. Conclusions

During a person’s lifetime, cells can accumulate different mutations as a result of aging, heritable and familial genetic factors, and exposure to physical, biological, and chemical substances [12,235]. The hallmarks of cancer, which identifies the series of actions and modifications where healthy cells can attain malignant characteristics, include de-differentiation (anaplasia), deregulated epigenetics, bypassing apoptosis and immunosurveillance, metabolic disarray, unrestrained proliferation, cellular quiescence, invasion, and metastases [192,236,237]. These hallmarks clearly show that tumorigenesis is a complex, heterogeneous process. Improvements in cancer diagnoses and treatments have advanced the ability to separate patients by risk and have made it possible to recommend therapy based on prognosis and patient preference [173]. Tumor subtype, anatomic cancer stage, and patient preferences are equally valuable for optimizing therapy for the individual patient [168]. Novel avenues of research are emerging due to the expansion of our knowledge of cancer hallmarks and understanding of BC and PC statistics related to diverse populations, population distribution patterns, and pathophysiology. We can be hopeful these investigations will improve patient outcomes and bring comfort to their families and friends. 

## Figures and Tables

**Figure 5 cancers-14-02954-f005:**
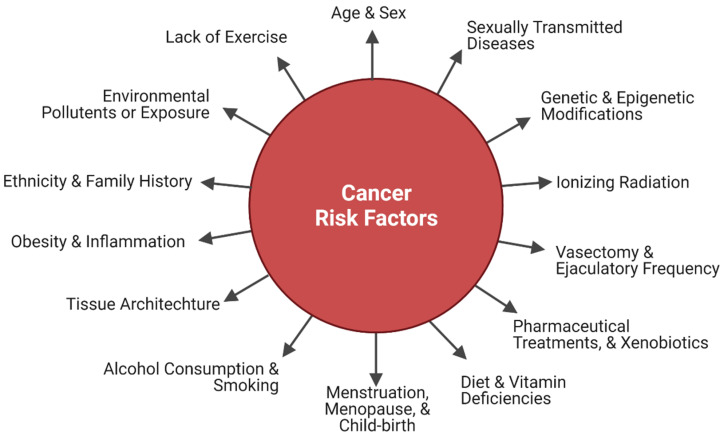
Cancer risk factors. There are multiple risk factors for breast and prostate cancer. Risk factors are either modifiable risk factors (e.g., diet, physical activity, and lifestyle related factors), or not (e.g., genetics, ethnicity, family history, and age).

## Data Availability

Not applicable.

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
