# Peer review of "A Tale of Two Cancers: A Current Concise Overview of Breast and Prostate Cancer"

_cancers, 2022, doi:10.3390/cancers14122954_

Round 1

Reviewer 1 Report

Dear Authors:

In the manuscript by Silva et al has summarized the current information on statistics, pathophysiology, risk factors, and treatments associated with breast and prostate cancer. I have just a few suggestions.

1. The references and information are missing:

In Page 2, Line 57-60:"GLOBOCAN 2020 reported deaths for female BC and PC as 6.9% and 3.8% respectively [8]. Accounting for all cancer sites, there was a cumulative risk of 22.6% (males) and 18.6% (females) for incidence, and a cumulative risk of 12.6% (males) and 8.9% (females) for mortality for both sexes (ages 0-74 years), globally [7,8,12,13]." In Page 3, Line 80-82:"BC is the most common form of malignancy among women in the developing and developed world [10,15,16]. One in 9 women in developed nations, and 1 in 20 in developing nations have a lifetime risk of developing BC [17]." Another review also demonstrated the epidemiology and pathogenesis of breast cancer.

In Page 6, Line 163-167:"In general, the breast is composed of adipose tissue, glandular tissue, and stromal (i.e., supporting) tissue in the breast parenchyma, the superficial fascia, and skin [43,44,51] (Fig 3). It is supplied by a web of blood vessels, lymphatic vessels and nodes, and nerves [21,52,53]. Approximately 97 % of the breast lymphatics drain to the axilla, while 3% drain to the internal mammary lymph nodes [44,53]." Another article also demonstrated the role of lymph node in breast cancer. 

In Page 10, Line 334-337:" A woman’s risk for developing BC increases with alcohol consumption and obesity where alcohol increases risk by 7% and an even higher frequency with every 10 g of daily alcohol consumption, while obesity increases risk of postmenopausal BC by 30 - 50% [110,112-114]." Another review also demonstrated the role of obesity in breast cancer.

In Page 13, Line 483-484:"Table 2. Currently available common treatment options for breast and prostate cancerous and non-cancerous conditions in North America. (Adapted from REF ‘cancer.ca’, ‘cancer.org’, and others[152-156]). There are more articles demonstrated developed and novel treatments for breast cancers. (please cite: 1. Patient Management Strategies in Perioperative, Intraoperative, and Postoperative Period in Breast Reconstruction With DIEP-Flap: Clinical Recommendations. Front Surg. 2022 Feb 15;9:729181. doi: 10.3389/fsurg.2022.729181. PMID: 35242802; PMCID: PMC8887567.

Best,

Author Response

Dear Reviewers,

We are grateful for all your suggestions, as well as for taking the time to peer-review our manuscript. We have verified the references and information and made some modifications to the text and references.

Based on your comments, we added further information along with a figure (Figure 6) on ‘drug-tolerant persister cells and relapse’ to the final section ‘Future Landscape’ of the manuscript.

All the modified components of the manuscript are highlighted.

Please feel free to let us know if there is anything else that can be done to improve our manuscript for publication.

Thank you very much!

Sincerely,

F. De Silva

Responses to Reviewer 1:

In the manuscript by Silva et al has summarized the current information on statistics, pathophysiology, risk factors, and treatments associated with breast and prostate cancer. I have just a few suggestions.

1. The references and information are missing:

In Page 2, Line 57-60:"GLOBOCAN 2020 reported deaths for female BC and PC as 6.9% and 3.8% respectively [8]. Accounting for all cancer sites, there was a cumulative risk of 22.6% (males) and 18.6% (females) for incidence, and a cumulative risk of 12.6% (males) and 8.9% (females) for mortality for both sexes (ages 0-74 years), globally [7,8,12,13]."

Response:

We have provided the correct references and information in the text which is highlighted on page 2 lines 57-62

2. In Page 3, Line 80-82:"BC is the most common form of malignancy among women in the developing and developed world [10,15,16]. One in 9 women in developed nations, and 1 in 20 in developing nations have a lifetime risk of developing BC [17]." Another review also demonstrated the epidemiology and pathogenesis of breast cancer.

Response:

We have provided the correct references and information in the text which is highlighted on page 3 lines 82-85

3. In Page 6, Line 163-167:"In general, the breast is composed of adipose tissue, glandular tissue, and stromal (i.e., supporting) tissue in the breast parenchyma, the superficial fascia, and skin [43,44,51] (Fig 3). It is supplied by a web of blood vessels, lymphatic vessels and nodes, and nerves [21,52,53]. Approximately 97 % of the breast lymphatics drain to the axilla, while 3% drain to the internal mammary lymph nodes [44,53]." Another article also demonstrated the role of lymph node in breast cancer. 

Response:

We added the correct references and information which is highlighted on page 6 lines 165-170

4. In Page 10, Line 334-337:" A woman’s risk for developing BC increases with alcohol consumption and obesity where alcohol increases risk by 7% and an even higher frequency with every 10 g of daily alcohol consumption, while obesity increases risk of postmenopausal BC by 30 - 50% [110,112-114]." Another review also demonstrated the role of obesity in breast cancer.

Response:

We added the correct references and information which is highlighted on page 10 lines 339-343

5. In Page 13, Line 483-484:"Table 2. Currently available common treatment options for breast and prostate cancerous and non-cancerous conditions in North America. (Adapted from REF ‘cancer.ca’, ‘cancer.org’, and others[152-156]). There are more articles demonstrated developed and novel treatments for breast cancers. (please cite: 1. . Front Surg. 2022 Feb 15;9:729181. doi: 10.3389/fsurg.2022.729181. PMID: 35242802; PMCID: PMC8887567.

Response:

We added the correct references and information in the table which is highlighted on pages 13-15.

Responses to Reviewer 2:

De Silva and Alcorn propose an article entitled “A Tale of Two Cancers: A Current Concise Overview of Breast and Prostate Cancer” for publication in MDPI-Cancers.

 In this review article, the authors discuss the various aspects of breast and prostate cancer and also discuss the available treatment strategies. The manuscript is straightforward, and concise within the scope of MDPI-Cancers.

However, I have a few minor concerns that should be addressed before publishing this work:

(1)   Minor grammatical mistakes were noted and should be addressed.

Response:

We reviewed the manuscript and made minor corrections throughout the text which are highlighted.

(2)   The principal reason for almost all the cancers including breast and prostate cancers are drug-tolerant persister cells which are responsible for relapse. The authors should at least discuss the salient features of these drug-tolerant persister cells. The authors are directed to PMID# 35584245 for this.

Response:

We have addressed this concept in the 'future landscape' section beginning on page 15 and is highlighted. We also included an additional figure (Figure 6) to address this concept. (Pgs 15-20).

Reviewer 2 Report

De Silva and Alcorn propose an article entitled “A Tale of Two Cancers: A Current Concise Overview of Breast and Prostate Cancer” for publication in MDPI-Cancers.

 In this review article, the authors discuss the various aspects of breast and prostate cancer and also discuss the available treatment strategies. The manuscript is straightforward, and concise within the scope of MDPI-Cancers.

However, I have a few minor concerns that should be addressed before publishing this work:

(1)   Minor grammatical mistakes were noted and should be addressed.

(2)   The principal reason for almost all the cancers including breast and prostate cancers are drug-tolerant persister cells which are responsible for relapse. The authors should at least discuss the salient features of these drug-tolerant persister cells. The authors are directed to PMID# 35584245 for this.

Author Response

(The authors gave the same response as above.)

Round 2

Reviewer 1 Report

Strongly suggest to publish.